

# Biopsy-based normalizations of gill monogenean-infected European catfish (*Silurus glanis* L., 1758) stocks for laboratory-based experiments

András Bognár[1], Muhammad Hafiz Borkhanuddin[2,3], Shion Nagase[1,4] and Boglárka Sellyei[3]

[1] Frontline Fish Genomics Research Group, Department of Applied Fish Biology, Institute of Aquaculture and Environmental Safety, Hungarian University of Agriculture and Life Sciences, Keszthely, Hungary
[2] Faculty of Science and Marine Environment, Universiti Malaysia Terengganu, Kuala Nerus, Terengganu, Malaysia
[3] HUN-REN Veterinary Medical Research Institute, Budapest, Hungary
[4] Graduate School of Marine Science and Technology, Tokyo University of Marine Science and Technology, Tokyo, Japan

Corresponding author
András Bognár,
bognar.andras@uni-mate.hu

## ABSTRACT

Ectoparasites cause serious problems during the aquaculture production of food fishes. In this study, we set out to develop and test protocols for maintenance and sampling European catfish (*Silurus glanis* L., 1758) stocks infected with a gill monogenean, *Thaparocleidus vistulensis* (Siwak 1932) Lim 1996. When we compared the feasibility of two cohabitation-based parasite culture systems (*i.e.*, static *vs.* flow-through), we found that the life cycle of *T. vistulensis* was completed in both habitats. In our experience, static tank systems with regular water exchange allowed better daily quality control of the parasite culture than continuous flow-through systems. We investigated the microhabitat preference of *T. vistulensis* on the gills of infected European catfish. A balanced distribution on the two lateral gill sets and a decreasing trend in parasite numbers from anterior gill holobranches towards the posterior ones was observed. Using these results, we developed a minimally invasive sampling protocol to estimate the parasite load of individuals. The biopsy aimed at four sectors (#6, #7, #10, and #11) situated within the distal and middle zones of the first holobranch on the left side, encompassing both rows of filaments. Biopsy-based estimates of parasite loads were validated by comparing them to full parasite counts of the same individuals and showed statistically significant correlations. Our biopsy-based method is designed to identify experimental animals with similar parasite loads and create groups of hosts with comparable burdens. This setup is expected to generate reduced between-group differences for expensive experiments (*e.g.*, high throughput transcriptomic or epigenetic studies). We propose that the biopsy-based pre-sorting procedure should be considered in similar experiments with other cultured fish species and their gill monogeneans following a thorough fine-tuning of the experimental conditions.

# INTRODUCTION

In natural ecosystems, parasites occur in relatively small quantities and exert comparatively minor influence on fishes at the population level, seldom causing mass mortalities (*Huntingford et al., 2006*; *Barber, 2007*). Parasite outbreaks in wild fish populations can be associated with anthropogenic activity: including fish-rearing systems that are in close contact with the surrounding aquatic environment and serve as disease hot spots; long-distance movement of live aquatic animals for farming; large-scale transport of animal products and inadequate aquaculture management (*Peeler & Feist, 2011*). Parasitic infections pose a continuous threat during the production of farmed fish species (*Hutson, Ernst & Whittington, 2007*; *Shinn et al., 2015*; *De Jesus et al., 2018*; *Overton et al., 2019*).

Parasite-loaded fish stocks are preferable over wild populations to collect infected specimens for expensive experiments, such as 'omics' studies using high-throughput technologies, or in cases where repeated sampling is needed. Several promising *in vitro* and *in vivo* culture techniques are under development for the propagation and maintenance of gill monogeneans in the laboratory (for a review see *e.g.*, *Hutson et al., 2018*). The permanent *in vivo* cultures of oviparous monogenean parasites could serve as a source of all life stages of the parasite for a prolonged time.

Monogeneans (Platyhelminthes, Monogenea) are diverse ectoparasites of marine, brackish and freshwater fishes, infecting their skin, gills or fins (*Poulin, 2002*; *Woo & Leatherland, 2006*; *Whittington & Chisholm, 2008*; *Hoai, 2020*). Monogeneans form a large clade with over a thousand species (*Poulin, 2002*). Moreover, they often show a microhabitat preference on the gill that seems to be a species-specific character. Gill monogeneans infecting siluriform fishes belong to the family Ancylodiscoididae, whose members typically possess two pairs of haptoral anchors and seven pairs of marginal hooks.

Most gill monogeneans—or gill flukes as they are often called in the aquaculture industry—such as species of the *Thaparocleidus* genus are host-specific. This means that they can infect only one or a few closely related fish species (*Poulin, 1992*; *Whittington et al., 2000*; *Mendlová & Šimková, 2014*) causing substantial losses in their farmed stocks, especially those kept in intensive systems (*Andree et al., 2015*; *Grano-Maldonado et al., 2018*; *Assane et al., 2022*). Farm-derived gill monogenean infections could have substantial negative impacts on the natural ecosystems through escaping carrier individuals (*Peeler et al., 2006*; *Mladineo et al., 2013*). Among the more than 150 known species of *Thaparocleidus* genus [WoRMS database (*Appeltans et al., 2012*)], three were reported (*Lim, Timofeeva & Gibson, 2001*) to cause specific infections on European catfish (*Silurus glanis* L. 1758). They are: *T. siluri* Zandt, 1924; *T. magnus* Bychowsky-Nagibina, 1957 and *T. vistulensis* Siwak, 1931, Lim 1996 (Common names and full Latin names of all fish and parasite species mentioned in the text will be listed in Table S1).

*Thaparocleidus* species infecting siluroids are small and can occur in high abundance (making their quantification rather time consuming). We looked for a method that provides a more accurate estimate of parasite infection intensity for *in vivo* treatments and selection experiments. In view of the above, this study had the following aims: (i) compare the suitability of two different rearing conditions (closed system without

recirculation *vs.* flow-through) for maintaining stocks of gill monogenean-infested catfish stocks; (ii) develop and characterize a gill biopsy-based method for rapid counting of the monogeneans; and (iii) test whether the biopsies could be used to estimate the total gill monogenean count of the infected catfishes to equalize the level of infection in groups for future experimental purposes. As a host-parasite model we used the European catfish and its most economically threatening ectoparasite *T. vistulensis* (*Molnár, 1968*). We report here on the adequacy of both systems to maintain the infected stocks, the improved method of infection by cohabitation and the distribution of *T. vistulensis* on the surface of gill holobranches. We show that the gill biopsy-based pre-testing allows adjustment of experimental groups with appropriate matching parasite levels for future use.

## MATERIALS & METHODS

### Origin and maintenance of catfish during the experiments

The experiments were conducted in two different laboratories. They are referred to as Experimental Location #1 (EL1); and Experimental Location #2 (EL2). In this study the experimental plans were reviewed and approved by the Hungarian National Scientific Ethical Committee on Animal Experimentation under the following ID numbers: GK-647/2020 (EL1) and PEI/001/1002-13/2015 (EL2).

At EL1, the European catfish (from here onwards catfish) individuals (size range: 48–84 g) used for the experiments originated from a genetically mixed stock. We used a closed tank system without water recirculation. An experimental catfish stock ($n = 40$) infected with *T. vistulensis* was established in two 170 L tanks, each containing 150 L of dechlorinated tap water and equipped with five XL size Bacto-Surge Foam Filters (Hikari, Japan). Approximately 30% of the total water volume was exchanged, excreta and other debris (possibly with parasite eggs) were removed daily to maintain well water quality. This aimed to keep the number of parasites below the critical level, where gill damage-induced hypoxia is fatal. Fish were fed dry commercial pellets of 2% total wet-fish body mass (Skretting Classic K 3P; Skretting, Stavanger, Norway) twice daily, and were kept at a temperature of $25 \pm 0.5$ °C with pH of 7, 13 $\pm$ 0.5. Oxygen saturation was kept above 80% while nitrate-ion concentrations in the systems typically ranged between 30–50 mg/L as determined separately under similar conditions. A catfish stock ($n = 30$) not showing signs of infection was reared in a separate tank under the same conditions to serve as potential replacements for dead individuals in the infected population.

At EL2, parasite-free catfishes from a genetically mixed stock weighing about 60–80 g were obtained from a pond of a commercial fish farm near Pécs (Hungary). The fish and the infected ones ($n = 25$) were kept in a continuous flow-through tank (50 L) system with a daily cumulative water exchange of 30% (1 L/h). During the trial period, activated carbon filtered tap water was used and the temperature was kept at 22 °C $\pm$ 2 °C and the average pH was 7.1 $\pm$ 0.4, with oxygen saturation maintained above 80% while nitrate-ion concentrations in the systems typically ranged between 30–50 mg/L as determined separately under similar conditions. Catfish were fed daily at 2% of their body weight with a commercial feed (Aller Bronze; Aller Aqua, Christiansfeld, Denmark). To keep the number of infected catfish constant, losses were replaced by naïve fish.

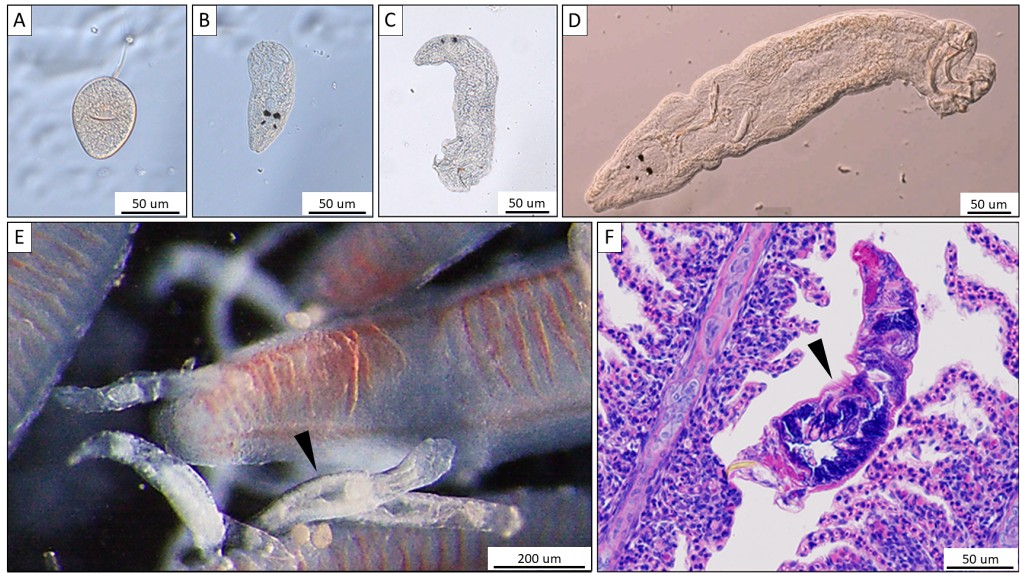

**Figure 1 Morphological characteristics of *Thaparocleidus vistulensis* observed under microscope.** (A–D) Developing stages of *T. vistulensis* recorded throughout the experiment: (A) single egg. (B–D) hatched oncomiracidium, juvenile and mature parasite with two pairs of eyespots, sclerotized copulatory organ and haptor with two pairs of hooks and seven pairs of marginal hooklets. (E) A parasite (arrowhead) attached to primary gill filaments of European catfish. (F) Hematoxylin-eosin-stained cross section of an adult parasite (arrowhead) revealing its inner anatomical features. Scale bars are 50 um (A–D; F) and 200 um (E), respectively.

## Examination and characterization of gill monogenean parasites

For the collection of monogeneans, catfish were sedated by immersing them into water containing 3–5 drops of clove oil (*Velíšek et al., 2006*) per 5 L aquarium water. Monogeneans were then removed from the gill or biopsy (see below) with a dissection needle and mounted on a slide under a coverslip. For identification and differentiation of the three potential *Thaparocleidus* species (*T. magnus*, *T. siluri*, and *T. vistulensis*), the number, shape and size of the hard parts of the haptor (anchors and hooklets) as well as the characteristics of sclerotized copulatory organ of the adult monogenean were observed (following *Gussev, 1962*; in *Bykhovskij & Pavllvskij (1962)* using an Olympus BH2 microscope at a magnification of × 400 and × 1,000. The developmental stages of *T. vistulensis* were recorded (Figs. 1A–1D) and specimens were preserved in 80% ethanol for subsequent analysis. Pathological alterations on the gill (Fig. 1F) were putatively induced *via* tissue disruption by *T. vistulensis* and potential secondary infections as published earlier (*Molnár et al., 2016*).

For histological analysis, pieces of gills infected with monogeneans were fixed in Bouin's solution (*Hughes, 1984*) to prevent shrinkage of lamellae and promote long-term storage (*Groff, 2001*). Fixed samples were embedded in paraffin wax, cut into 4–5 μm sections, and stained with hematoxylin and eosin. The histological slides were observed and photographed with an Olympus BH2 microscope equipped with an Olympus DP 20 digital camera.

## Experimental infection of catfish by co-habitation

The infection status of catfish reared in the intensive system was estimated by observing the (piece of) gills (Fig. 1E) under a stereo microscope. Hereafter, catfish with a low total number of parasites (<200; for details see the next section) will be called 'normal', and individuals with more than 700 gill monogenean counts—'heavily infected' according to the prediction of pathogen-loads.

At EL1, ten heavily infected catfish individuals were placed into a 10L floating cage to cohabitate with 40 normal individuals in a 150 L tank. A foam filter was used inside the tanks instead of external filters, to avoid continuous parasite egg removal (by filtration) from the water. At EL2, 5–6 heavily infected catfish individuals were kept in a floating cage (5 L) immersed into a 50 L cohabitation tank containing 20 normal individuals to ensure their infection.

At both experimental locations, the water was aerated slowly through air stones with continuous flow of bubbles. This ensured the recirculation of water and parasite eggs. Moreover, it allowed the fish to keep their buoyant position without active swimming, at the same time. The tanks were decorated with stones and artificial plants to mimic natural conditions. Tanks were inspected twice daily at least, and moribund or dead individuals were removed. The persistence of *T. vistulensis* infection and the absence of other - unwanted -parasites were validated continuously by checking the gills of individuals removed. The floating cage with heavily infected individuals was removed from the tank after two weeks.

## Estimation of parasite load by a minimally invasive method

At EL1, infected catfish individuals were anaesthetized by adding 3–5 drops of clove oil in five liters of aquarium water (*Velíšek et al., 2006*). Then the operculum was lifted and a $3 \times 3$ mm incision was made using surgical scissors equipped with curved blades. (Fig. S1). The biopsy included four sectors on gill filaments (#6, #7, #10 and #11) located in the distal and middle zones according to the clustering system proposed by *Lo & Morand (2000)*. Since each gill arch includes two rows of filaments, thus the biopsy consisted of two pieces (Fig. S1). The almond-shaped incision never reached proximal zone of the filaments. All equipment and surfaces involved in the biopsy were disinfected through flaming prior to each use. Following the biopsy, fish were transferred into a separate tank containing well-aerated water to regain their consciousness and full locomotor activity. The parasite count obtained from the biopsy allowed the estimation of each individual's total parasite load. All catfish that underwent the procedure and fulfilled the expectation about pathogen burden based on the biopsy, marked by clipping the dorsal fin.

Validation of the estimates was performed by comparing the biopsy-based gill monogenean count with a total monogenean load. Biopsies were conducted as described above. To assess the total monogenean load of a single catfish, it was euthanized using an overdose of clove oil. After the postmortem examination, all monogeneans were counted from the eight dissected gill holobranches (four on each side).

## Generation of catfish groups by incidental collection, using biopsy-based parasite load and hypothetical pairwise-matched pairing

At EL2, the smaller (<70 gr) and the larger (>70 gr) fish were separated to avoid size related parasite number bias, then splitted with a single netting motion, incidentally picking individuals after cohabitation, with no apparent changes in behavior. There was no effort to match parasite loads among the groups. (Fig. 2, Experimental location 2) At EL1, following the completion of the cohabitation-based infection process, experimental groups were created using biopsy parasite numbers. Based on the parasite count, batches B1 and B4 were formed from individuals that were infected with less than 10 ($n < 10$); in group B3 with less than 20 ($n < 20$) monogeneans; whereas group B2 was composed of individuals with at least a two-digit parasite number in the biopsy. To model biopsy-based experimental group formation, we created two hypothetical experimental groups using the biopsy parasite numbers of individuals from B1–B4 groups: catfish with a similar monogenean load in the biopsy were matched and paired *in silico*, and the two members of the couple were eventually divided between the two hypothetical experimental populations (Fig. 2, Experimental location 1). None of the individuals used in these experimental groups showed clinical signs resulting from the infection.

### Statistical analysis

Monogenean counts on the left *vs.* right branchial basket were compared with a Paired *t*-test after passing normality tests (D'Agostino & Pearson; Kolmogorov–Smirnov) (*Lilliefors, 1967*; *D'agostino & Pearson, 1973*). A two-tailed *p*-value was accepted as significant, if $p < 0.05$. Effectiveness of pairing was $r = 0.9183$. Pairing was significantly effective: one-tailed $p < 0.0001$.

Gill holobranch preference of parasites of the same catfish was measured four times (once for every holobranch) on the same dependent variable (parasite number). The data was organized vertically into four different categories (B1-B2-B3-B4 as holobranches) but horizontally belongs to the same individual (related groups). Analysis was performed with Friedman test, as a non-parametric alternative to one-way ANOVA with repeated measures, since the dataset did not meet the requirements of normal data distribution (D'Agostino & Pearson; Kolmogorov–Smirnov). Significant value of means was accepted if $p < 0.05$. To identify pairs whose members differed significantly between groups, Dunn's multiple comparison test was applied with adjusted *p*-values, accepted as significant, if $p < 0.05$.

The strength of the linear relationship between a pair of variables (total parasite count–biopsy parasite number) was shown with correlation analysis. Dotted lines represent error lines. Among the 25 displayed individuals, four were out of range and three were on the error line, representing a 27% deviated group of fish. During the analysis we computed r for *X versus* every *Y* dataset. Pearson correlation coefficients were calculated with a 95% confidence interval for *p*-value (two-tailed). Spearman correlation coefficient if $r \geq 0.6$ was accepted as a strong positive correlation between the two datasets. Calculated *p*-value (two-tailed) for the Pearson correlation coefficient was stated as significant if $p < 0.0001$.
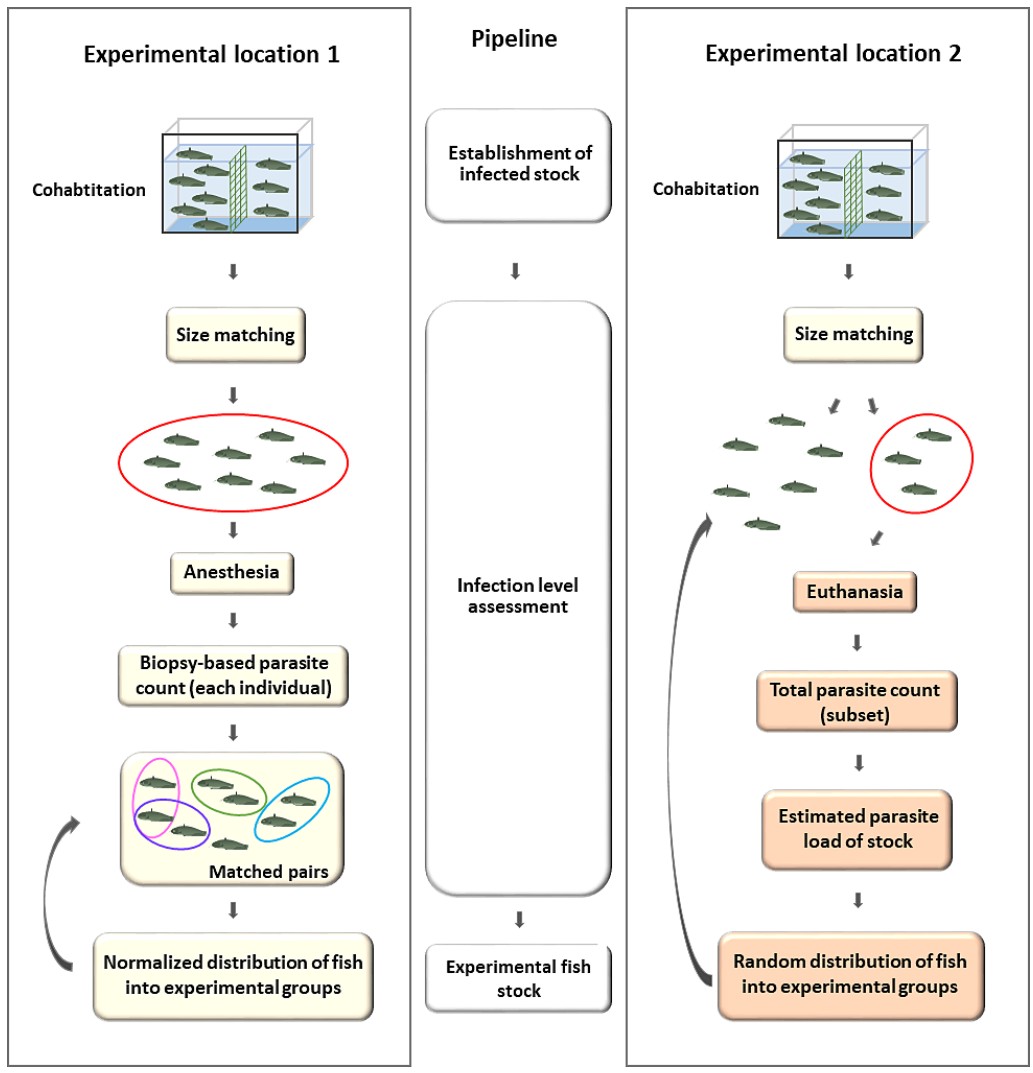

**Figure 2** Generation of experimental groups of catfish infected by cohabitation, using a biopsy-based estimation of parasite load (EL1) and incidental collection (EL2). EL1 (left): Infection was achieved by cohabitating infected fish with size-matched normal behaving individuals. After the infection period, biopsy-based sampling of the first gill holobranch on the left side was performed under anesthesia to match tagged hosts based on parasite counts. Pairs of hosts with similar parasite loads were formed and then used for further experimental purposes, by placing members of a pair into two different groups (curved arrow). EL2 (right): Infected catfish population was formed by cohabitation (see above). The infection level of the stock was estimated based on total parasite counts of a few incidentally collected, sacrificed hosts, whereas the rest of the fish was used for the experiment (curved arrow). There was no attempt to match parasite loads between the treated and control groups. The components of the figure are original artwork.

To test whether parasite numbers show equal population variances between experimental groups (EL1; EL2), the Brown-Forsythe test was used due to the non-Gaussian nature of our datasets (*Glantz, Slinker & Neilands, 2001*). The assumption of equal variances was violated, if $p < 0.05$.

Analysis of significant difference between the two hypothetical paired experimental groups means was performed by Wilcoxon matched-pairs signed rank test as a non-parametric alternative to paired $t$-test. Difference was accepted as significant if $p < 0.05$. Justification of the usage of the paired test was validated by the significant effectiveness of pairing ($p = 0.0019$) and Spearman rs $= 0.7811$.

All *T. vistulensis* records are shown in Table S2. Statistical data analysis and visualization were performed using GraphPad Prism version 8.0 for Windows (GraphPad Software, La Jolla, CA, USA).

## RESULTS

### Static and flow-through tank systems are both suitable alternatives for lab-based maintenance of catfish infected with *T. vistulensis*

We tested the maintenance of gill monogenean-infected catfish individuals in two different rearing systems: one with static conditions including in-tank filtration (EL1) and one with a continuous flow through system (EL2). Both systems operated with a total daily exchange of 30% water volume. Our data showed that 20–40 infected catfish individuals can be kept under these two conditions. We managed to maintain the parasite infection in all the groups throughout the experimental periods, and none of the groups suffered such a large-scale loss of fish (exceeding 25%) that could interfere with the subsequent statistical analysis of data. The different life stages of *T. vistulensis* (Wan Sajiri et al., 2023) could be observed on the gills (Figs. 1A–1D), confirming the appropriate conditions for parasite reproduction in both EL1 and EL2.

### The spatial distribution of parasites was even on both sides of the host gills but showed a preference for the first two anterior gills

We aimed to investigate whether *T. vistulensis* countings by a gill biopsy could be used for the estimation of the overall load of the catfish individuals to achieve a more homogenous parasite burdened group of hosts at EL1. As previous studies had shown differences in the spatial distribution of parasites on the gills of teleosts (see *e.g.*, discussed) we analyzed this issue to determine the optimal position for biopsy.

First, we compared parasite counts on all four holobranches located on the gills of left-*versus* right side of 12 infected catfish individuals at EL1. The combined parasite number on the left gill basket was not significantly different from that of the right side ($p = 0.683$; $n = 12$; Fig. 3A).

Then, we analyzed the spatial distribution of *T. vistulensis* among the four gill holobranches on the left side. Data from 21 catfish individuals showed that there was no significant difference between the number of monogeneans on the first and second anterior holobranches ($p = 0.384$; Fig. 3B). On the other hand, the first two anterior holobranches (*i.e.,* those two located closer to the operculum) contained a significantly higher number of parasites, than the third and fourth ($p < 0.05$ for both; Fig. 3B).

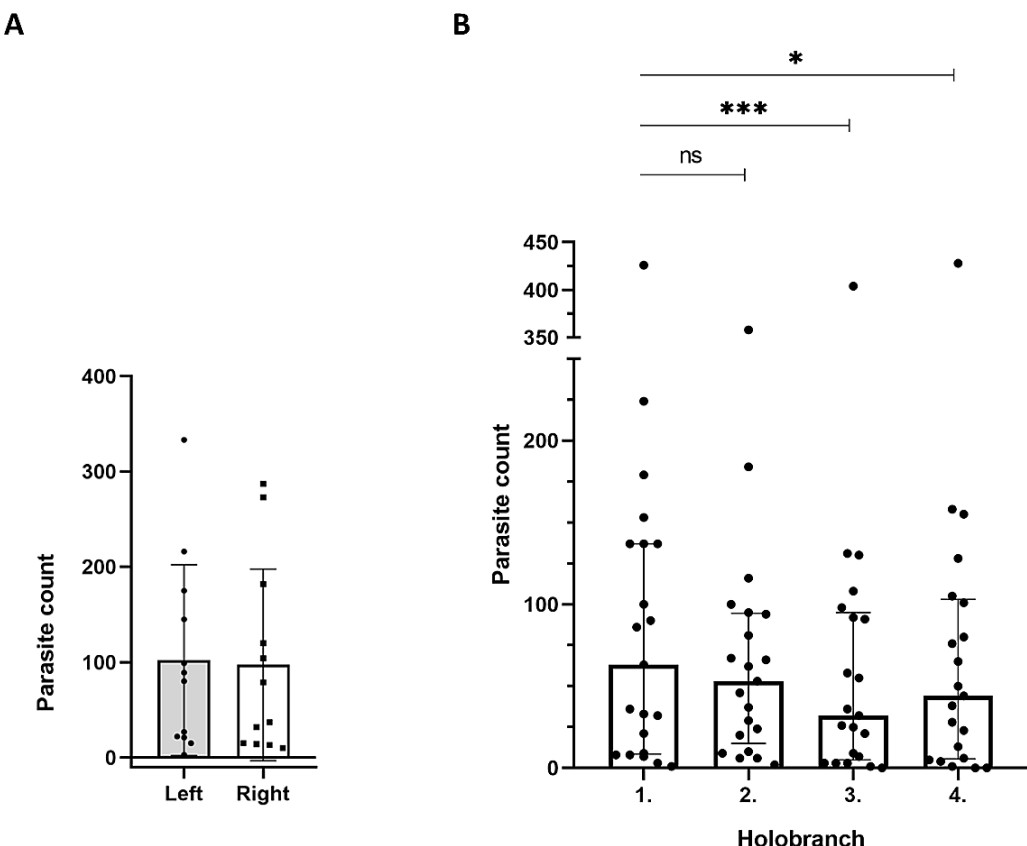

**Figure 3** **Spatial distribution patterns of the *Thaparocleidus vistulensis* parasite on the gills of catfish.** (A) Equal distribution of parasites between the left and right side of gills. The distribution of parasites was not significantly different (paired $t$-test) between left (grey column; $n = 12$) and right (white column; $n = 12$) lateral gill holobranches (error bars, mean $\pm$ SD; two-sample $t$-test; $p = 0.683$). (B) Parasite loads of the first two anterior gill holobranches were significantly higher than those of the third and fourth, respectively. When pairwise comparisons of parasite count of the four holobranches on the left side were performed, there was no significant difference between the counts on the first and second gill holobranches ($n = 21$; Friedman test with Dunn's multiple comparison test; $p = 0.384$). However, the third and fourth gill holobranches contained significantly lower number of parasites, than any of the first two ($p < 0.05$ for both). The total parasite number for each holobranch is represented by a dot. The individual on the top had an exceptionally high parasite load (note the broken $Y$ axis), but it displayed a similar trend to the rest.

### Biopsy-based gill monogenean count is a proper tool for estimating the overall parasite load

The parasite counts in the biopsy taken from the first anterior holobranch from the left side of each individual showed a significant correlation ($p < 0.001$; Spearman $r = 0.7913$) with the total parasite count on the entire gill set of the same host (Fig. 4). The biopsy yielded a good estimation for the level of infection in almost 90% of the 25 individuals tested, only four individuals deviated from the trend.

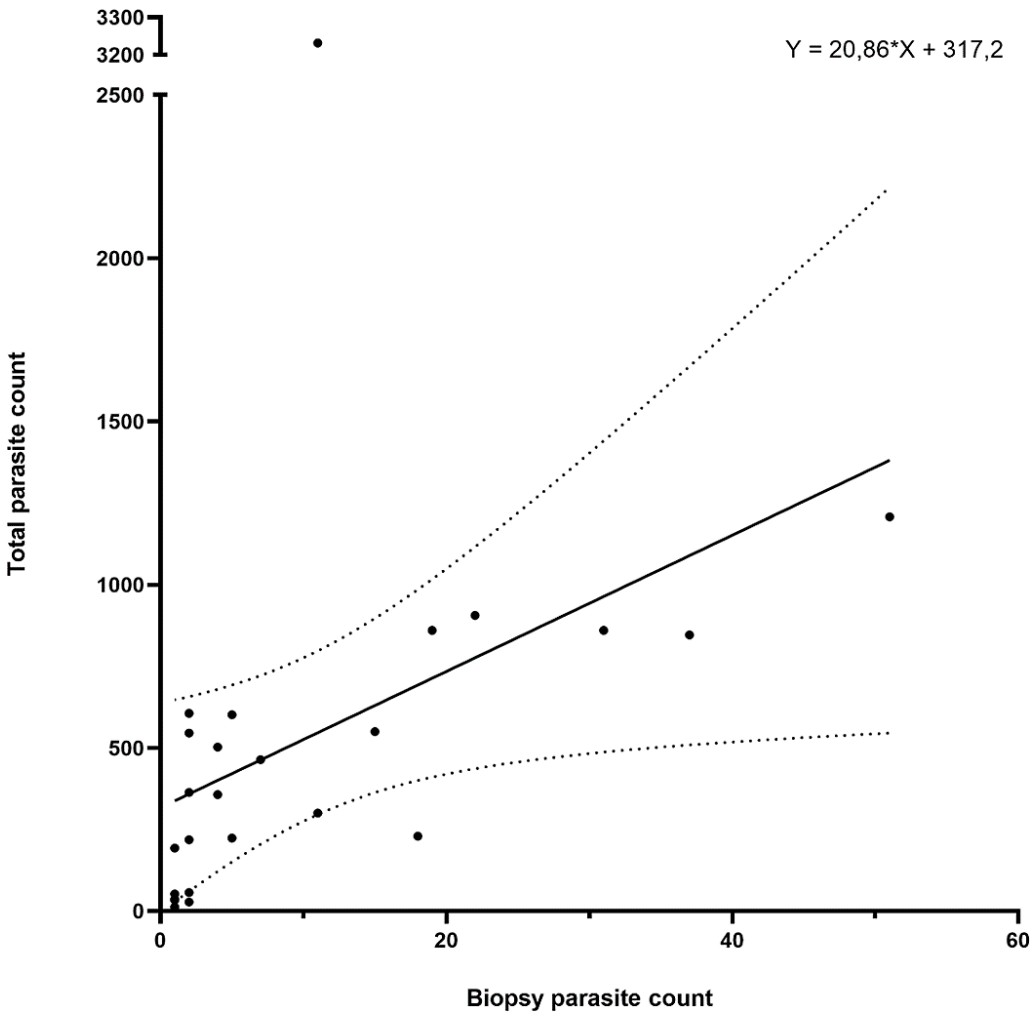

**Figure 4** **Gill biopsy-based parasite counts show statistically significant correlation with total parasite counts from the same individual.** Data from 25 individuals are shown. Each point represents an individual, the correlation trend is depicted as a continuous line, while error bars are connected with dotted lines. Correlation: $p < 0.001$; Spearman $r = 0.7913$.

## Biopsy-based sorting allows for lower parasite number variance within batches of infected catfish

Individuals ($n = 25$) were collected incidentally at EL2 to analyze the differences in the total parasite load among hosts. Our analyses showed that the assumption of equal variances had been violated for total parasite numbers between EL2 experimental catfish groups ($p = 0.001$; Fig. 5A), yielding groups with heterogeneous total parasite numbers among individuals.

At EL1, we generated groups using individuals available at a time with very similar biopsy monogenean parasite loads, to test whether using this method would result in catfish groups with equal population variances of parasite counts. Total parasite numbers

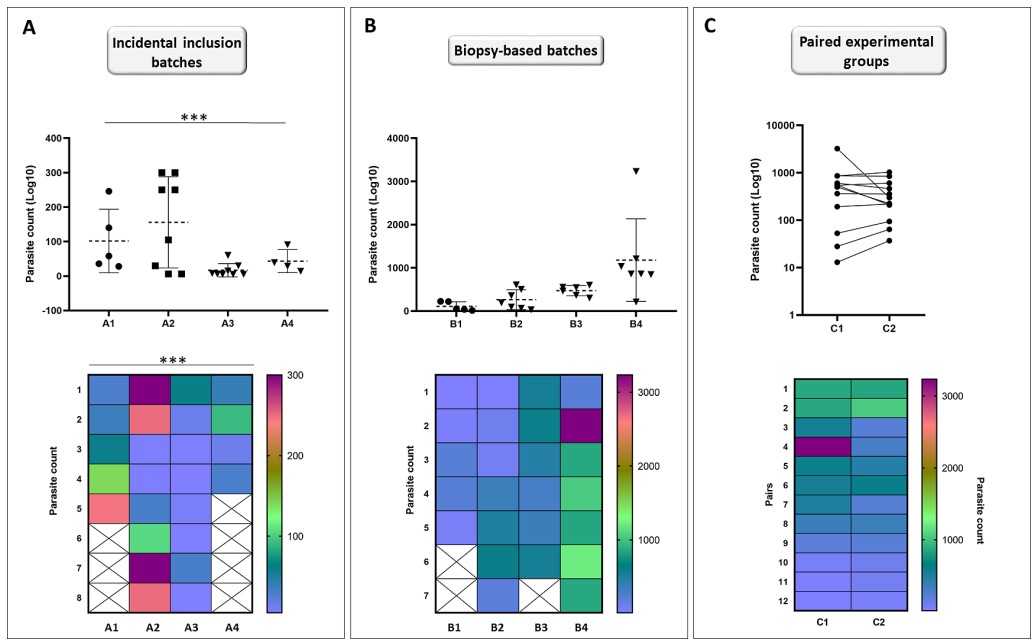

**Figure 5** **Incidental collection of catfish leads to high variation within groups while biopsy-based sorting allows for similar population variances; and formation of hypothetical matched pairs using biopsy parasite counts leads to normalized experimental population.** Catfish batches generated by incidental sampling (A, batches A1–A4) showed significant differences (Brown-Forsythe test) between the SD ranges among the four batches ($p = 0.001$). Variation level among the total parasite counts of four independent batches (B, batches B1–B4) generated after biopsy-based sorting were highly similar (Brown-Forsythe test, equal population variances: $p = 0.350$). Points represent individuals on the scatter dot plots, dashed lines indicate mean ± SD. Hypothetical pairwise-matched experimental groups (C, groups C1–C2) generated using data from the biopsy-based batches (B, batches B1–B4) showed no difference with Wilcoxon matched-pairs signed rank test ($p = 0.7334$). Number of individuals per group is shown in Table S1. On the symbols and lines plot, individuals are shown as points and pairs are represented as connected points. Rainbow heat maps are visual demonstrations of parasite count variances of batches and hypothetical groups.

of the four EL1 biopsy-based experimental groups showed equal population variances ($p = 0.350$; Fig. 5B).

Then using the same biopsy parasite counts, we created two hypothetical pairwise matched experimental groups (Fig. 5C). Wilcoxon matched-pairs signed rank test showed no significant difference ($p = 0.7334$) between the total parasite numbers of the matched pairs, validating the biopsy-based pairwise assembly grouping method and the effective pairing ($p = 0.0019$) of *T. vistulensis* infected experimental European catfish.

These results demonstrate the benefit of the biopsy-based formation of experimental groups of individuals with similar infection ranges for future experiments.

## DISCUSSION

*Thaparocleidus vistulensis* is a monogenean ectoparasite on the gills that causes serious problems for the aquaculture of European catfish, leading mass losses in the stocks. Our aim is to understand better this host-parasite interaction, hoping to develop effective

treatments for infected catfishes. For that, we needed to maintain catfish stocks infected with *T. vistulensis* under laboratory conditions, and established laboratory procedure for sorting infected catfish with a similar overall load for the experimental groups.

In this paper, we tested two different captive managements for tanks to allow the maintenance of a gill monogenean-infected catfish stock. Both (EL1and EL2) were based on the same cohabitation method: normal behaving recipients and infected hosts are kept physically separated in the same water body (Fig. 2). This setting allows parasites to infect new hosts and complete their life cycle. A 30% water change in EL1, and continuous flow-through of water with a 30% daily cumulative change rate at EL2 remove most free-swimming larvae, and thereby prevent the excessive proliferation of monogeneans. Studies on fish pathogens can inevitably lead to the loss of hosts (*Stephens et al., 2003*; *Hernández-Cabanyero et al., 2023*); the removal of heavily infected catfish and replacement with new, uninfected individuals facilitate the long-term maintenance period but might contribute to evolving differences in parasite counts. The equipment and footprint of the required infrastructure are relatively small. The rearing systems presented are suitable for small-scale laboratory experiments and maintenance of infection, even by inexperienced persons. The static aquarium system (EL1) is more flexible and allows fine control of water temperature. As the in-model systems complete reproductive cycle of monogeneans tend to last for 8–11 days at a temperature of 20 °C (*Zhang et al., 2015*), the ascending tendency of infection status of the stock can be expected after two weeks at the temperature range of 23−25 °C. After four weeks of the introduction of the infection, the first check of parasite load and assessments can be started.

Substantial variations in the parasite loads of individual hosts occur even in well-controlled culture systems. The reasons for these could be genetically controlled (*e.g.*, different sensitivity of individuals) or random (*e.g.*, former injuries, differences in activity, replacement of heavily infected individuals by naive ones, *etc.*). Accordingly, the resistance, survival and parasite load of animals in the same stock may vary within wide scales (*Magnadottir, 2010*). Omics-based studies will likely yield different results from the tissues of hosts with highly variable levels of parasite load. Forwood and colleagues (*2012*) developed an improved method for estimating the intensity of the monogenean parasite, *Lepidotrema bidyana*, on the posterior hemibranch (L1p) of the left gill basket in 25 individual silver perch (*Bidyanus bidyanus*). A robust correlation was demonstrated between the posterior hemibranch (L1p) and the overall parasite counts intensity. This method is well-suited for predicting individual-level parasite burden *in situ* (*Forwood, Harris & Deveney, 2012*) and post-treatment parasite abundance (*Forwood, Harris & Deveney, 2013*). The method accurately estimates parasite intensity and abundance; however, the removal of the posterior (L1p) hemibranch can only be performed *post mortem*, after euthanasia.

In a different experiment, gill biopsies were isolated from the ventral region of the second gill holobranch of randomly selected West Australian dhufish (*Glaucosoma hebraicum*) to determine the infestation level of the monogenean, *Haliotrema abaddon* (*Stephens et al., 2003*). Moderate and heavy infection level of stock was estimated using biopsy parasite numbers, however without correlating with total counts. Subsequent comparative

treatment trials and effective monogenean removal was evaluated from gill biopsies of fish without prior gill sampling of individuals (*Stephens et al., 2003*).

In view of these considerations, a preliminary assessment of the individual parasite load is highly recommended for experiments involving targeted treatment or aiming to discover the molecular mechanisms of parasite-host interaction using high-throughput technologies. Gill biopsy is routinely used in teleosts to study physiological responses, including ion transport (*Mancera & McCormick, 2000*; *Pelis, Zydlewski & McCormick, 2001*; *McCormick et al., 2008*), immune function (*Yada, McCormick & Hyodo, 2012*) and microbiome (*Clinton et al., 2021*), as well as veterinary health assessments (*Hadfield, 2021*; *Novotny, 2021*; *Seeley, 2021*). We adopted a gill biopsy-based method for the estimation of the level of *T. vistulensis* infection in European catfish. The procedure is minimally invasive and easy to perform. Throughout the experimental work described in this study, none of the individuals showed long-term negative effects caused by the biopsy. Others have also used gill biopsy to verify the number of monogeneans in greater amberjack (*Seriola dumerili*) (*Rigos et al., 2021*), as well as to detect the presence and intensity of parasites in dhufish (*Stephens et al., 2003*) and European catfish (*Khara & Sattari, 2016*). According to our knowledge, this study is the first to demonstrate a correlation between biopsy-based *T. vistulensis* monogenean counts and the total number of parasites present on the gills of infected individuals in the latter species.

The bilateral distribution of monogeneans between the left *vs.* right gill sides was analyzed to identify the optimal site for sample collection and to test whether the area of biopsy represents properly the total parasite number of the entire set of gills. In most previous studies, the difference in the microhabitat preference of monogeneans between the right and left gill sides in teleosts was not commonly reported (*Tombi, Nack & Bilong, 2010*; *Kumar, Madhavi & Sailaja, 2017*; *Zolovs, Kanto & Jakubāne, 2018*). The four exceptions were as follows: right side dominance were shown for *Dactylogirus amphibothrium* and *Microcotyle mugilis* (*Wootten, 1974*; *Wootten, 1974*); and for *Gyrodactylus salaris* (*Peeler et al., 2006*); while left side preference was detected in *Metamicrocotyle cephalus* (*El Hafidi et al., 1998*). Our data indicate that *T. vistulensis* showed no side preference in European catfish - we have chosen the left side for subsequent biopsy collection.

Gill holobranches and spatial locations within could be loaded with different numbers of the same monogenean species due to the microhabitat preference of the parasite. To determine whether such preference exists for *T. vistulensis* was necessary to determine which gill holobranch should be targeted for the biopsy. Comparing the parasite load of the four gill holobranches on one side, we detected a marked difference. The first and second gill arch filaments contained significantly more parasites, than the third and fourth—this is the first observation of the unequal distribution of *T. vistulensis* among the gill holobranches of European catfish. The reasons for this difference are not known. We cannot exclude the possibility that this distribution pattern was affected by the biopsy sampling, but we think that this outcome is less likely, given the facts that the parasite is tightly connected to the tissue with several hooks and the relatively short duration of the analysis of the four holobranches. Microhabitat preference among the gill holobranches on one side was reported earlier for four out of eight ectoparasites (*Lamellodiscus major, Polyabroides*

*multispinosus, Lernanthropus atrox* and *Clavellopsis parasargi)* tested on yellowfin bream (*Acanthopagrus australis*) (*Roubal, 1982*) and *Zeuxapta seriolae* on yellowtail kingfish (*Sharp et al., 2003*).

*Lo & Morand (2000)* studied the spatial distribution of parasites on the gill of two damselfish species (*Stegastes nigricans* and *Dascyllus aruanus*) and developed a detailed system for the division of the gills into 24 sections. We have not checked the potential differences among gill sections of European catfish, nonetheless adapted their system to keep the location of biopsy reproducible. We have chosen filaments from the first arch as the target for the biopsy. Our data indicate that the gill biopsy-based correlation analysis can be used to obtain reasonable correlations for the overall load of catfish individuals infected with 200 to 1,000 monogeneans.

Biopsy analysis can provide an assessment of the infection level for the entire stock and identify fish with similar parasite loads for further use. If these paired individuals are used to form the control and treatment groups, they will have a very similar average parasite load and range, so the interpretation and reliability of potential comparative experiments will be easier and more reliable *via* the reduction of the measurement noise during the analysis. Our simplified, yet effective culture system and the biopsy-based estimation of total parasite counts offer potential tools for those researchers, who intend to conduct gill monogenean-based experiments focusing on the host in the future. Since gill parasites can exhibit microhabitat specificity with different intensities of specific regions (*Roubal, 1982*), the proposed method must be validated for each model. We expect that the biopsy-based estimation method will be easily adaptable for estimating overall loads of other monogeneans that specifically infect gills of different teleost species, yet likely require extensive testing, adaptation, and final validation.

## CONCLUSIONS

Both static and flow-through tank systems are suitable alternatives for lab-based establishment and maintenance of *T. vistulensis*- infected experimental stocks of European catfish. A non-lethal gill biopsy taken from a gill holobranch is a reliable tool to estimate the parasite load of an individual host in the size range of the examined catfish. Our observations of the consistent bilateral distribution of *T. vistulensis,* and the anterior-posterior decreasing tendency in the parasite number among holobranches on the left side allowed the identification of the proper site for biopsy isolation.

Matched pairs or batches assembled from biopsy-analyzed fish showed no statistical differences between variances in total parasite numbers. Consequently, using gill biopsy-based grouping of experimental European catfish at the analyzed size range could be used to decrease bias caused by different parasite loads in comparative therapeutic investigations.

## ACKNOWLEDGEMENTS

The authors would like to thank Dr. László Orbán, Dr. Ildikó Szeverényi and Dr. Tamás Molnár for their help during the preparation and revision of the manuscript. They are

grateful to Dr. Kate Hutson and two anonymous referees for their helpful criticisms and suggestions that have contributed the revision of the manuscript.

### Funding

András Bognár was supported by the Frontline Research Grant of the National Research, Development and Innovation Office of Hungary (KKP 140353). Muhammad Hafiz Borkhanuddin recieved support from the Ministry of Higher Education, Malaysia (MOHE) for the MHB Postdoctoral Program. Shion Nagase received financial support of the Tobitate! (Leap for Tomorrow) Study Abroad Initiative by the Ministry of Education, Culture, Sports, Science and Technology, Japan, and the support of Prof. Carlos Strüssmann with the arrangement of her study trip to Hungary. The funders had no role in study design, data collection and analysis, decision to publish, or preparation of the manuscript.

### Grant Disclosures

The following grant information was disclosed by the authors:
Frontline Research Grant of the National Research, Development and Innovation Office of Hungary: KKP 140353.
Ministry of Higher Education, Malaysia (MOHE).
Ministry of Education, Culture, Sports, Science and Technology, Japan.

### Competing Interests

The authors declare there are no competing interests.

### Author Contributions

- András Bognár conceived and designed the experiments, performed the experiments, analyzed the data, prepared figures and/or tables, authored or reviewed drafts of the article, and approved the final draft.
- Muhammad Hafiz Borkhanuddin performed the experiments, authored or reviewed drafts of the article, and approved the final draft.
- Shion Nagase performed the experiments, prepared figures and/or tables, and approved the final draft.
- Boglárka Sellyei performed the experiments, authored or reviewed drafts of the article, and approved the final draft.

### Animal Ethics

The following information was supplied relating to ethical approvals (i.e., approving body and any reference numbers):
Hungarian National Scientific Ethical Committee on Animal Experimentation

### Data Availability

The raw data is available in the Supplemental Files.

## Supplemental Information

Supplemental information for this article can be found online at http://dx.doi.org/10.7717/peerj.18288#supplemental-information.

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
