# Peer review of "Biopsy-based normalizations of gill monogenean-infected European catfish (Silurus glanis L., 1758) stocks for laboratory-based experiments"

_PeerJ, doi:10.7717/peerj.18288_

## Round 0.1 · original submission · Major Revisions

Please see all the comments from the reviewers, including the annotated drafts. Please address all the comments.

PLEASE PAY SPECIAL ATTENTION TO THE CONCERNS REGARDING ANIMAL WELFARE

**Language Note:** The review process has identified that the English language must be improved. PeerJ can provide language editing services - please contact us at [email protected] for pricing (be sure to provide your manuscript number and title). Alternatively, you should make your own arrangements to improve the language quality and provide details in your response letter. – PeerJ Staff

Reviewer 1 ·

Basic reporting

There are some issues with the English:
- non-sentence adverbs should not be used at the beginning of sentences (e.g. "However" and "Here" and "Hence" and particularly "Furthermore") because this use makes the reference for the adverb unclear because they do not limit or describe the meaning of the whole sentence to which they refer
- modifying adverbs such as "extremely" should not be used because they can't be qualified or quantified - provide information i.e. "The number of known monogeneans is over 10,000" or similar
- do not use the term "believe" - science is based on evidence. "Data support that" or another term that refers to support for an idea must be used
- the abstract needs to provide some more information about the approaches used rather than merely be descriptive - for example there is no mention of what statistics or approach are used, or how the MS assesses the data
- references following scientific names need to be differentiated from the species authorities (e.g. Hutson et al. are not the authority for Seriola lalandi and Rigos et al. are not the authority for Seriola dumerili)
- authorities are used inconsistently - please provide authorities for all species names
- Taxa higher than genus are not italicized (see the ICZN for guidance)


There are some basic proofreading issues:
- line 62 has a full stop/period before the reference Peeler & Feist that needs to be removed
- line 68 "most of monogeneans" should be "most monogeneans"
- the term "decimate" means to kill one in ten. It is inaccurate on line 81
- Hutson reference on line 99 needs reformatting
- line 106 "parasitic" should be "parasite"

Background
- site specificity and the need for this work is inadequately explained and needs to be clarified

Experimental design

The following need to be clarified:
- what you aimed to do and why is not properly explained
- the design shown in Figure 2 is not described in the Materials and Methods
- line 124 what is "a critical level"?
- line 126 does "healthy" mean "uninfected by T. vistulensis"? Clarify
- line 147 - Bouin's solution is not normally used for fish tissue histology? Why was it used? How was the picric acid removed from the sections?
- line 157 what size are the floating cages?
- line 162 what does "slowly aerated" mean?

Validity of the findings

- parasite populations generally do not conform to the assumptions of the Friedman test. How was this test chosen? How were the pairwise gill arch comparisons set up?
- how did you assess the "normal" variance in the infections? Did you use power analysis to ensure that the sample sizes used in the studies were adequate to account for the variation?
- If the data were not normal, how was the Brown-Forsyth test chosen, given that it assumes a normal distribution?
- how were the correlations assessed?
- The results do not explain what you found - the amount of deviation in non-conforming fish needs to be described
- line 252-3 is a statement of achievement for an aim that has never been properly articulated

Additional comments

I am concerned about the use of death as an endpoint by parasite infection described from line 166. Was there ethical approval for this work? There is no ethics statement, which is a journal publication requirement.

Is there regulatory approval for the use of clove oil? Methyl eugenol is clove oil is carcinogenic, and its use has been discouraged in most jurisdictions

The first 2 paragraphs of the discussion are largely repetition of the Materials and Methods or Results. Delete or edit to make relevant here

Reviewer 2 ·

Basic reporting

Some basic editing of the English language is needed before acceptance. Article structure and references are acceptable. Discussion based on hypothesis and results.

This is an interesting manuscript describing the establishment of an in vivo system for producing a single-species monogenean infection on catfish. There are a number of edits and comments on the pdf draft of the manuscript that the authors should consider and clarify before it is acceptable for publication.

Additional concerns are as follows:
1. Ethical use of animal statement or IACUC approval number not listed in Methods.
2. There is a significant animal welfare concern in this manuscript with their description of the infection process. For instance, in lines 155 and 166, the authors state “abnormal behaviours: loss of light avoidance; erratic swimming (towards water surface) and gasping for air and gathering at the outlets of tanks due to their reduced swimming ability as a result of hypoxia” and “Dead and apathetic catfish individuals with complete loss of motor function and floating uncontrollably”. All of these clinical signs/conditions are past the acceptable humane endpoint and present unacceptable animal care for fish in captivity, and even in the catfish used as a source of infection presents unacceptable fish welfare.
3. Line 215. The authors mention that “none of the groups suffered large-scale loss of fish”. What do the authors consider large-scale? 50%, 75% or even 90% loses. The authors need to clarify this point, but again it still presents unacceptable animal welfare.
4. The authors need to discuss that this technique may only work with Thaparocleidus species as other species of monogeneans may only infect the skin, or the skin and gills.
5. The authors do not state the desired number of monogeneans per catfish desired for experimental infections, and catfish showing clinical signs are not suitable as research animals.
6. Lines 175 and 334, the authors use the term “lamellae” incorrectly as a gill biopsy contains several filaments which are composed of numerous lamellae. What the authors are describing is taking a biopsy of one of the hemibranchs which contains a set of filaments coming off the gill cartilage (arch).
7. Line 173. Does sedation of the fish with clove oil remove any of the monogeneans from the gills? Did the authors check the bottom of the sedation chamber for detached worms?
8. Figure 1F – there is significant gill pathology occurring in the image. The authors need to discuss whether this was cause by the parasite, poor water quality, or other factors.
9. Figure 4A and 4B – these graphs to not support the authors’ statement in Line 335 that “analysis showed that catfish individuals with a total parasite load between 200 and 1000 are in the most reliable prediction range”. It looks like 0 to 100 would be the only predictive range.

Experimental design

This is original research with a focused hypothesis. Additional revision and edits of Methods needed.

Validity of the findings

Discussion based on collected data, but may be a bit overstated in conclusions.

Additional comments

Biggest concern is the ethical treatment of the animals and the lack of ethical statement in manuscript Methods section.

Annotated reviews are not available for download in order to protect the identity of reviewers who chose to remain anonymous.

·

Basic reporting

- Suggest to improve English expression
- Suggest to include additional literature
- Professional structure for articles needs to be carefully followed; there are new methods presented in the results.
-Not all of the raw data were shared (e.g. parasite counts from each gill arch)

Experimental design

- Yes, research question is valid and reasonably well defined
- Methods do not have sufficient detail

Validity of the findings

- Rationale needs to include a statement on validation for future models
- Appears to be underlying data missing

Additional comments

Please see attached a detailed review document

---

## Round 0.2 · Minor Revisions

Thank you for addressing the comments from the reviewers. This significantly improved the manuscript, however as the reviewer states, there are still some issues which need to be addressed. In particular, we still have concerns about animal welfare in this study. Please clarify what you mean by maintaining infection with "no major losses" as number of percentage and how this can be justified by animal welfare. Please include water quality data, including ammonia and pH as the minimum.

Additional comments:
- please explain what random selection procedures you used for the fish selection, remembering that random in statistics means that each fish had equal chance to be selected. This is very hard to do in fish sampling and it is important that the method is clarified or the word "random" or "randomly" should not be used.
- please use "tissue" in its scientific meaning and not colloquial. Gills are an organ, they are composed of various tissues. So it is best to say "sample" instead of "tissue sample" or "two pieces" instead of "two tissue pieces".
Please make sure that all the comments from the reviewer are addressed.

Reviewer 2 ·

Basic reporting

This is a much improved and more concise manuscript. I still have concerns about the ethical treatment of the animals (i.e. no animal should be allowed to show clinical signs or suffer) but at least the authors have now included their animal experimentation ethics approval. In addition, the authors present very little water quality data other than temperature and dissolved oxygen (i.e. they should have also monitored ammonia, nitrites, nitrates and pH for example). This lack of water quality data is unacceptable in most peer-reviewed journals any more, thus both of these concerns will need to be dealt with in future research and manuscripts. There are minor English edits on pages 5, 12, 15 and 17 of pdf.

Some additional comments the authors should correct.
Lines 211, 267 and 332 - what is a branchial “basket”? Use “cavity” or “chamber”.
Lines 358-360 - This statement may be true for this manuscript, but would not be true for all other species on monogeneans. The authors should clarify this since not all species of monogeneans occur on both the gills and body, some occur on only the gills while others only occur on the body. This will need to be determined for any particular species used as a model for infectious studies.
Line 368-369 - have the authors thought this may be a result of which gill arches were collected first, i.e. left side up or eight side up first. Depending on the time required to collect gill biopsies from one side may allow time for the monogeneans on the other side to redistribute.

Experimental design

I'm still concerned about some of the ethical treatment of the animals. This research probably would not have been approved if the authors were using rats, mice, dogs of cats. Yes, fish are vertebrate animals too!

Validity of the findings

Some of the conclusions are a bit overstated, and should pertain to only this manuscript.. I'm also still concerned that the authors did not monitor water quality as it should be in a research situation.

Additional comments

There are minor English edits on pages 5, 12, 15 and 17 of pdf.

Annotated reviews are not available for download in order to protect the identity of reviewers who chose to remain anonymous.

---

## Round 0.3 · accepted · Accept

Thank you for addressing all of the reviewer's comments. I have assessed the revision myself and I am happy with the current version. The manuscript is ready for publication.